# DEFNTAXS: THE INEVITABLE NEED FOR CONTEXT IN CLASSIFICATION

## ABSTRACT

To successfully use generalized vision-language models (VLMs) like CLIP for zero-shot image classification, the semantics of the target classes must be well defined and easily differentiated. However, test datasets rarely meet either criterion, implicitly encoding ambiguity in labels, even when adding class-specific descriptors. Existing literature focuses on improving text inputs by using class-specific descriptors to further refine taxonomic granularity, but largely fails to leverage higher-order semantic relationships among classes. We introduce Defined Taxonomic Stratification (DefNTaxS): a fully automated, procedural, training-free framework that leverages large language models (LLMs) to cluster related classes into hierarchical subcategories and augment CLIP prompts with this taxonomic context. By sculpting text prompts to boost both semantic content and inter-class differentiability, DefNTaxS disambiguates semantically similar classes and improves classification accuracy. Across seven standard benchmarks, including ImageNet and EuroSAT, DefNTaxS achieves up to 13.0% absolute accuracy gain (average 5.5%) over vanilla ViT-B/32 CLIP and consistent improvement over other recent SOTA, all while enhancing semantic interpretability without any model retraining/modification, manual prompt alteration, or additional optimization data.

## 1 INTRODUCTION

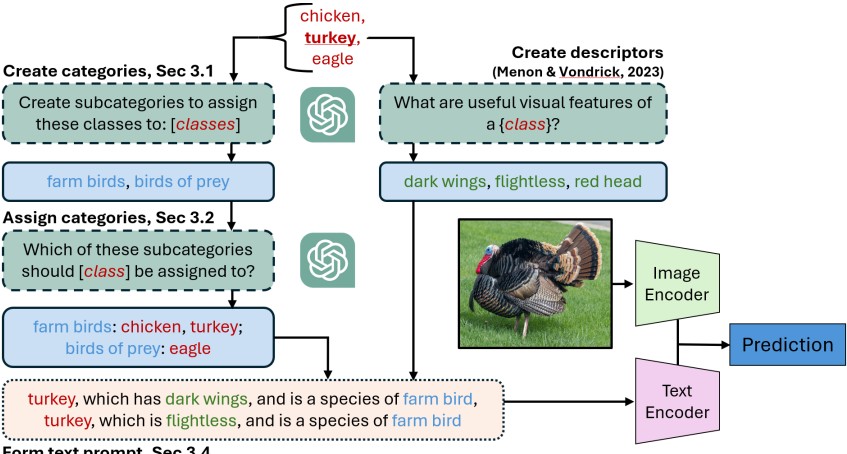

Figure 1: While D-CLIP (Menon & Vondrick, 2023) is limited to fine-grained visual descriptors, DefNTaxS builds a comprehensive semantic framework for each class, e.g. "turkey".

The power of vision-language models like CLIP (Radford et al., 2021) lies in their ability to perform zero-shot image classification by matching visual content with textual descriptions. However, a critical limitation emerges when class labels are ambiguous: "boxer" appears in datasets as both a dog breed and a fighting sport, "crane" as both a bird and construction equipment, and "mouse" as both an animal and computer peripheral.

Current approaches tackle this challenge through two main strategies: **descriptive enhancement** and **hierarchical refinement**. Descriptive methods like D-CLIP (Menon & Vondrick, 2023) add visual features ("boxer which has short fur"), while hierarchical approaches like CHiLS (Novack et al., 2023) organize classes into tree structures. Yet both strategies have fundamental limitations that prevent them from solving the core ambiguity problem.

Humans effortlessly resolve ambiguity by leveraging taxonomic context - we understand "boxer" differently when it appears among "dog breeds" versus "combat sports." This taxonomic disambiguation is absent from current zero-shot classification methods, which treat each class in isolation or focus only on hierarchical relationships without considering lateral semantic groupings that aid differentiation.

Existing methods fall short in three critical ways: **(1) Contextual blindness**: Descriptive approaches like D-CLIP focus on isolated features ("has short fur") without considering how classes relate to each other within the dataset, missing the taxonomic context that resolves ambiguity. **(2) Incomplete disambiguation**: While CHiLS uses hierarchical structures, it doesn't leverage lateral relationships between classes - crucial for distinguishing "boxer dog" from "boxer athlete" when both could be described similarly. **(3) Semantic isolation**: Current methods treat each class independently, ignoring that classification confidence also often comes from understanding what a class is *not*, relative to other classes in the same domain.

**Our Solution: DefNTaxS (Defined Taxonomic Stratification)** We introduce a framework that automatically discovers taxonomic context and integrates it into vision-language model prompts. DefNTaxS uses LLMs to analyze datasets and discover meaningful subcategories, then enriches prompts with both descriptors and taxonomic context (e.g., "boxer, which has a muscular build, commonly found among dog breeds").

**Key Contributions: (1) Taxonomic Discovery Algorithm**: We develop an automated method for discovering task-relevant taxonomies that maximize inter-class differentiation in zero-shot settings. **(2) Disambiguation Through Context**: We demonstrate that taxonomic context is not just helpful but *essential* for robust zero-shot classification, addressing fundamental ambiguity issues overlooked by existing methods. **(3) Strong Empirical Results**: DefNTaxS achieves consistent improvements across seven benchmarks (+5.5% average, +13.0% maximum gain over CLIP), establishing new state-of-the-art results without requiring model retraining or manual intervention.

## 2 RELATED WORK

**Vision-Language Models for Zero-Shot Classification** VLMs like CLIP (Radford et al., 2021) learn joint text-image representations, enabling zero-shot classification by matching image embeddings with textual prompt embeddings Yuan et al. (2021); Yu et al. (2022); Singh et al. (2022); Yao et al. (2022); Li et al. (2022). These models are foundational to generalizing contrastive alignment methods to unseen data and tasks (Cho et al., 2021; Zhang et al., 2023).

**Prompt Engineering in VLMs** Radford et al. also introduced prompt engineering through E-CLIP: manually crafting 80 prompt templates to improve zero-shot classification accuracy, e.g. "a photo of a [class]". Prompt engineering is now common practice for structuring a model's inputs. However, as prompt optimization requires extensive manual effort Radford et al. (2021), automated prompt tuning methods have been developed to efficiently learn prompt representations (Zhou et al., 2022).

**LLM Prompt Augmentation** Recent works leverage LLMs (Brown, 2020) to source text prompts (Menon & Vondrick, 2023; Roth et al., 2023; Pratt et al., 2023). D-CLIP Menon & Vondrick (2023) pioneered this direction, appending discriminative LLM-generated visual descriptors to class labels, e.g. appending "which has white wings" to the "Albatross" class label. WaffleCLIP Roth et al. (2023) swaps these descriptors with random characters for similar outcomes, suggesting that VLMs cannot process fine-grained semantics, but high-level concepts impact inter-class differentiability. CuPL (Pratt et al., 2023) generates free-form prompts, granting nuance at the cost of consistency and control. Existing LLM-driven methods target narrow aspects of the nature of a class, neglecting other elements that may reduce ambiguity and enrich semantic context.

**Taxonomy and Hierarchy-Based Methods.** Beyond descriptor-based approaches, taxonomy- and hierarchy-based methods focus on inter-class relationships. CHiLS (Novack et al., 2023) constructs

hierarchical label sets by generating or refining class hyponyms as proxies for base class classification. Ren et al. (2024) propose a ChatGPT-powered hierarchical comparison framework that organizes classes into multi-level taxonomies and aggregates across a multi-level fused scoring system, demonstrating robustness on benchmarks with high inter-class similarity. However, these methods treat each hierarchical level independently without directly integrating cross-level context, and are also inapplicable to fine-grained datasets due to limitations in granularity.

**DefNTaxS** addresses the limitations of these methods by integrating multiple taxonomic levels into a unified prompt. Our approach combines the strengths of LLM-generated descriptors and taxonomy-based methods, ensuring each class is comprehensively defined within the task-relevant context.

## 3 METHOD

DefNTaxS addresses the fundamental ambiguity problem in zero-shot classification through **taxonomic contextualization**, automatically discovering semantic relationships between classes and integrating this knowledge into enhanced prompts.

**Core Insight**: Effective disambiguation requires understanding not just what a class *is*, but what domain it belongs to and how it differs from related classes. A "boxer" becomes unambiguous when understood as "boxer, which has a muscular build, commonly found among dog breeds" rather than simply "boxer which is muscular."

DefNTaxS operates through four steps: **(1) Taxonomic Discovery**, **(2) Contextual Assignment**, **(3) Granularity Optimization**, and **(4) Prompt Enhancement**, transforming isolated labels into contextually rich prompts (Figure 1).

### 3.1 TAXONOMIC DISCOVERY: AUTOMATED SUBCATEGORY GENERATION

Given a dataset with $n$ classes, DefNTaxS leverages large language models to discover $k$ meaningful subcategories that capture essential semantic relationships. Unlike hierarchical approaches that impose rigid tree structures, our method identifies **lateral semantic groupings** that maximize disambiguation potential.

We prompt the LLM to analyze all classes simultaneously, identifying natural clusters based on shared properties and contextual domains (e.g., distinguishing "water birds" from "birds of prey"). Our approach enforces clean partitioning:

1. **Complete coverage:** Every class belongs to exactly one subcategory
2. **Non-overlapping assignment:** No class appears in multiple subcategories
3. **Semantic coherence:** Each subcategory represents a meaningful semantic domain

Formally, this creates a partition $\{s_1, s_2, \ldots, s_k\}$ of the class set $C$ where:

$$\bigcup_{i=1}^{k} s_i = C \quad \text{and} \quad s_i \cap s_j = \emptyset \quad \text{for all} \quad i \neq j \tag{1}$$

This partitioning ensures that each class gains taxonomic identity without conflicting assignments, providing the foundation for effective disambiguation.

### 3.2 STEP 2: ASSIGNING CLASSES TO SUBCATEGORIES

After generating the subcategories, we assign each class to its most appropriate subcategory. This process uses efficient parallel LLM calls, where each call determines the best subcategory for a single class (detailed prompting strategies are provided in Appendix Appendix A).

The assignment considers the context of all classes in the dataset, leading to intelligent, context-aware groupings. For instance:

- If a dataset contains "boxer," "boxing gloves," and "tennis racket," then "boxer" would likely be assigned to a "sports" subcategory

- If the same dataset contains "boxer," "bulldog," and "poodle," then "boxer" would more likely be assigned to a "dogs" subcategory

- In the edge case the dataset contains classes of both sports and dogs, we run a check for the class already assigned to a subcategory and instruct the LLM to avoid that subcategory when assigning the class again, looping until a unique assignment is found

This contextual assignment ensures that each class gains semantic meaning not just from its own properties, but from its relationships with other classes in the same subcategory.

## 3.3 STEP 3: REFINING SUBCATEGORIES

The size and specificity of subcategories significantly impacts classification performance. Through empirical analysis (Section Appendix D), we determined that approximately 20 classes per subcategory yields optimal results. This guides our refinement process to avoid two common problems:

1. **Overly broad subcategories:** Groups like "animals" or "objects" are too general to provide meaningful differentiation between classes

2. **Overly specific subcategories:** Groups with only one or two classes fail to provide useful context, as the LLM will often name the subcategory after the single class it contains, e.g. "the croissant subcategory" for a subcategory containing only "croissant"

**Ensuring sufficient subcategories:** We require at least $\frac{n}{20}$ subcategories for $n$ classes. If the LLM generates too few subcategories, we prompt it to create more specific groups. For example, "dogs" might become "small dogs," "medium dogs," and "large dogs."

**Splitting large subcategories:** Any subcategory with more than 20 classes is automatically split into smaller, more focused groups through additional LLM refinement.

**Handling small datasets:** For datasets with fewer than 20 classes, we use the dataset name as the single subcategory context (e.g., "EuroSAT dataset"), as creating multiple subcategories may harm performance.

**Technical controls:** We limit LLM token output to prevent incomplete responses and implement string processing to ensure coherent subcategory lists.

## 3.4 STEP 4: CREATING CONTEXTUAL DESCRIPTIONS

Rather than simply appending subcategory names to class labels, we generate natural contextual phrases that describe how classes within each subcategory relate to one another. This makes the final prompts more informative and linguistically natural.

For each subcategory, the LLM creates a connecting phrase that captures the semantic relationship among the grouped classes. These phrases vary based on the nature of the subcategory:

- For "kitchen utensils": "commonly found among"

- For "dog breeds": "a type of"

- For "vehicles": "used for transportation as"

The result is natural-sounding prompts like:

- "forks, which have tines, commonly found among kitchen utensils"

- "golden retriever, which has long fur, a type of dog"

- "sedan, which has four doors, used for transportation as a car"

This contextualization provides richer semantic information than simple concatenation, helping the vision-language model better understand class relationships and distinctions.

## 3.5 BUILDING THE FINAL TEXT PROMPTS AND CLASSIFICATION

The final step combines all generated components into comprehensive text prompts for classification. Each prompt integrates three key elements:

1. **Class name**: The original class label (e.g., "fork")
2. **Descriptive features**: Class-specific attributes generated similar to D-CLIP Menon & Vondrick (2023) (e.g., "has tines")
3. **Taxonomic context**: The subcategory context phrase (e.g., "commonly found among kitchen utensils")

These elements are combined following the template:
"*[class]* which *[has/is] [descriptor]*, *[contextual phrase] [subcategory]*"

Example result: "fork, which has tines, commonly found among kitchen utensils."

**Classification Process:** To classify a query image, we (1) generate multiple text prompts for each class using consistent subcategories but different descriptors, (2) compute similarity scores between the image embedding and each text prompt embedding, (3) average the scores across all prompts for each class, and (4) select the class with the highest average score.

Mathematically:

$$\text{Score}(c) = \frac{1}{|D_c|} \sum_{d \in D_c} \text{similarity}(\text{image}(x), \text{text}(\text{prompt}(c, d))) \tag{2}$$

$$\text{Predicted class} = \arg\max_c \text{Score}(c) \tag{3}$$

where $D_c$ is the set of descriptors for class $c$.

## 4 EXPERIMENTAL SETTINGS

### 4.1 IMPLEMENTATION/EVALUATION DETAILS

Unless specified otherwise, all experiments are conducted on a single RTX 4090 GPU. The descriptors used in the experiments are generated using a modified version of D-CLIP's generation pipeline (Menon & Vondrick, 2023) due to the deprecation of OpenAI's GPT-3 API (Brown, 2020). The classification accuracy is reported as the primary evaluation metric in a pure zero-shot setting on each dataset's standard training split.

### 4.2 DATASETS

For evaluating our method, we use the benchmark outlines provided in Menon & Vondrick (2023) for zero-shot classification. This benchmark consists of ImageNet (IN) (Deng et al., 2009), a dataset for classifying everyday objects; CUB (Welinder et al., 2010), which focuses on fine-grained bird species classification; Oxford Pets (Pets) (Parkhi et al., 2012), for the recognition of common pets; DTD (Cimpoi et al., 2014), used for texture classification in natural settings; Food101 (Food) (Bossard et al., 2014), aimed at food categorization; Places365 (Places) (Zhou et al., 2017), a large-scale dataset for scene and environment recognition; and EuroSAT (ESAT) (Helber et al., 2017), which focuses on land use and land cover classification based on Sentinel-2 satellite imagery.

We used GPT-4o-mini for all experiments, with total text generation cost of $0.38 USD.

### 4.3 BASELINES

We compare the performance of DefNTaxS against several SOTA methods for zero-shot image classification using VLMs. The baselines and their prompt formats include:

- **CLIP** (Radford et al., 2021), of the format "$\{class\}$".

- **E-CLIP** (Radford et al., 2021) uses hand-crafted templates such as "a photo of $\{class\}$".

- **D-CLIP** (Menon & Vondrick, 2023), which generates class-specific descriptors using LLMs of the format "$\{class\}$ which has/is $\{descriptor\}$".

- **WaffleCLIP / W-CLIP** (Roth et al., 2023) replaces LLM-generated descriptors with random characters, of the format "a photo of a $\{class\}$ which has/is $\{random\ words/characters\}$".

- **WaffleCLIP + Conc.** (Roth et al., 2023) includes a high-level semantic concept generated by GPT-3 (Brown, 2020) into the WaffleCLIP format, "a photo of a $\{concept\}$, a $\{class\}$, which has/is $\{random\ words/characters\}$".

- **CuPL** (Pratt et al., 2023) generates multiple free-form prompts for each class to capture the nuances of each category, with no specific format.

- **CGPT-P** (Ren et al., 2024) uses a hierarchical tree search classification approach, where hierarchical labels are generated by ChatGPT, fusing the classification scores from various hierarchical levels rather than a score from a single prompt.

- **CHiLS** (Novack et al., 2023) leverages either established or LLM-generated hyponyms in place of base class labels, mapping hyponym predictions to their base classes.

Each baseline was recreated using the setup described in 4.1 and the code provided for each study. All potential variables were maintained strictly to those used in the original studies, and in doing so, controlled for any inconsistencies due to hardware, software, or other implementation variables.

## 5 RESULTS

DefNTaxS demonstrates that taxonomic context is not merely helpful but **essential** for robust zero-shot classification. Our thorough evaluation across seven standard benchmarks reveals consistent and substantial improvements.

| | IN | CUB | Pets | DTD | Food | Places | ESAT | INV2 | Mean |
|---|---|---|---|---|---|---|---|---|---|
| CLIP | 58.89 | 51.86 | 77.88 | 41.12 | 77.83 | 37.50 | 44.26 | 51.70 | 55.13 |
| E-CLIP | 61.90 | 52.00 | 82.10 | 43.07 | 78.78 | 39.13 | 33.44 | 52.73 | 55.40 |
| D-CLIP | 63.00 | 53.21 | 81.84 | 43.62 | 80.43 | 39.84 | 47.36 | 55.77 | 58.13 |
| W-CLIP | 62.35 | 51.73 | 82.38 | 40.05 | 79.43 | 38.35 | 31.49 | 52.98 | 54.85 |
| W-CLIP+conc. | 62.35 | 52.73 | 85.40 | 40.05 | 81.25 | 40.22 | 40.81 | 53.27 | 57.03 |
| CUPL | 62.12 | 52.34 | 81.78 | 40.95 | 79.84 | 38.87 | 41.50 | 52.88 | 56.29 |
| CGPT-P | 63.32 | 53.72 | 82.90 | 45.81 | 80.98 | 39.91 | 43.11 | 54.31 | 57.97 |
| CHiLS | 60.94 | 51.98 | 85.73 | 43.67 | **83.53** | **40.45** | 42.83 | 53.55 | 57.74 |
| *DefNTaxS* | **63.48** | **54.00** | **86.09** | **45.89** | 81.48 | 40.00 | **57.22** | **56.43** | **61.17** |
| $\rightharpoonup \Delta$ CLIP | +4.59 | +2.14 | +8.21 | +4.77 | +3.65 | +2.51 | +12.96 | +4.73 | +5.44 |
| $\rightharpoonup \Delta$ D-CLIP | +0.48 | +0.79 | +4.25 | +2.27 | +1.05 | +0.16 | +9.86 | +0.66 | +2.44 |

Table 1: Comparison of DefNTaxS zero-shot classification performance against standard benchmarks, and both descriptor-based and taxonomic baselines.

Table 1 shows DefNTaxS achieving the highest accuracy across six of seven benchmarks, with an average improvement of +5.5% over vanilla CLIP and consistent gains over recent methods. Even where other baselines maintain SOTA, DefNTaxS maintains strong performance, usually outperforming the third place by a reasonable margin.

The most compelling results emerge on datasets with high semantic ambiguity. EuroSAT shows a remarkable +13.0% improvement (57.22% accuracy), where taxonomic context helps distinguish land use categories that share visual similarities but differ in satellite imagery context. Similarly, ImageNet benefits significantly from disambiguating the hundreds of potentially ambiguous classes.

DefNTaxS maintains strong performance on ImageNetV2, demonstrating that taxonomic context provides genuine disambiguation rather than dataset-specific overfitting. The consistent improvements across diverse domains - from fine-grained bird classification (CUB) to texture recognition

(DTD) - establish taxonomic contextualization as a fundamental advancement in zero-shot classification.

Beyond accuracy gains, DefNTaxS offers significant practical advantages: completely automated operation, no additional training data requirements, and seamless integration with existing CLIP models. The total computational cost for generating all text across all datasets is less than $0.40, making this approach immediately deployable for real-world applications.

## 6 ABLATION

The DefNTaxS approach focuses on the addition of taxonomic semantic content and the subsequent contextualization of fine-grained semantics in text prompts. However, other important aspects of these prompts have not been controlled for. In the following sections, we further explore the effects of both the taxonomic semantic content and the differentiation of semantic content in text prompts.

### 6.1 VARIATION OF CORE METHOD ELEMENTS

Existing research analyses the effects of semantic descriptors as discriminators at various hierarchical depths, with some studies focusing on the benefits of fine-grained descriptors (Menon & Vondrick, 2023). Other literature focuses on the advantages of high-level semantic concepts (Roth et al., 2023), while asserting that VLMs struggle to leverage the actual semantics introduced through fine-grained semantic descriptors. Here we investigate the real impact of semantic descriptors at various hierarchical depths on zero-shot classification performance.

### 6.1.1 REDUCED TAXONOMIC REFINEMENT

| Method | IN | Places |
|---|---|---|
| CLIP | 58.86 | 37.48 |
| E-CLIP | 61.90 | 39.12 |
| D-CLIP | **63.26** | **40.89** |
| W-CLIP | 60.25 | 38.28 |
| W-CLIP+Conc. | 60.25 | 38.28 |
| DefNTaxS | 61.23 | 37.53 |

Table 2: Effect of reduced taxonomic refinement on zero-shot classification performance.

For larger datasets, especially ImageNet and Places365 with hundreds of classes each, the taxonomic assignment process may disproportionately allocate a large number of classes to a single subcategory, reducing the ability of these subcategories to increase the differentiation between classes. For example, a subcategory like "dogs" may contain all of the 150+ different dog species in ImageNet. To investigate the impact of this scenario, we limit the taxonomic refinement process to a single iteration and remove the refinement stages. The results of this investigation shown in Table 2 demonstrate a significant decrease in classification accuracy. As with other descriptor-based methods, even if the additional per-class semantic context is increased, the lack of differentiation between classes damages the ability of the VLM to distinguish between them. This highlights the importance of the taxonomic refinement process in DefNTaxS, which ensures that each subcategory is sufficiently distinct through a manageable number of subcategories.

### 6.1.2 DEFNTAXS WITH MODIFIED DESCRIPTORS AND SUBCATEGORIES

| Method | IN | CUB | Pets | DTD | Food | Places | ESAT |
|---|---|---|---|---|---|---|---|
| DefNTaxS | **63.48** | **54.00** | **86.09** | **45.89** | 81.26 | **40.00** | **57.22** |
| → tax. desc. | 59.80 | 51.21 | 85.66 | 41.26 | 79.29 | 35.67 | 51.22 |
| → no desc. | 62.62 | 53.76 | 85.64 | 44.49 | **81.35** | 38.56 | 55.90 |

Table 3: We modify the DefNTaxS structure by either **adding taxonomic descriptors** (tax. desc, e.g. "a croissant, which has flaky pastry and is found on a menu under pastries**, which are often sweet**") or all descriptors removed (no desc, e.g. "a croissant, found on a menu under pastries").

Research on hierarchical approaches to zero-shot text-image classification largely focuses on either ascending or descending levels of descriptive resolution, but rarely both. DefNTaxS leverages the

benefits of greater taxonomic semantic content from all levels of the hierarchy, from the class level to the subcategory level. However, it is not always clear how much of the performance improvement is due to semantic content relative to other factors.

To add further semantic content, we generate taxonomic subcategory descriptors, allowing us to directly compare the relative benefits of semantic content across hierarchical depth (3). We explore the impact due to both the inclusion of subcategory descriptors and removal of class descriptors.

We observe strong performance of the standard DefNTaxS approach relative to both modifications. While the addition of taxonomic subcategory descriptors reducing performance may suggest that the additional semantic content is not beneficial, further investigation would be needed to determine the cause of this performance drop. The context window of CLIP is limited to 77 tokens, much more than the 20-30 tokens used in the DefNTaxS prompts, but literature suggests that effective context window size is much closer to 20 tokens (Zhang et al., 2024), potentially impacting this result. However, despite the addition of semantic content having unclear impact, we have also observed that increased taxonomic context without fine-grained differentiation can damage performance.

### 6.1.3 DIFFERENTIATION WITHOUT SEMANTIC CONTENT

| Method | IN | CUB | Pets | DTD | Food | Places | ESAT |
|---|---|---|---|---|---|---|---|
| DefNTaxS | $62.96 \pm 0.26$ | $\mathbf{53.59 \pm 0.20}$ | $\mathbf{86.17 \pm 0.07}$ | $\mathbf{45.83 \pm 0.63}$ | $\mathbf{81.10 \pm 0.09}$ | $39.34 \pm 0.26$ | $\mathbf{55.99 \pm 0.36}$ |
| W-TaxS | $\mathbf{63.24 \pm 0.06}$ | $\mathbf{53.65 \pm 0.17}$ | $84.49 \pm 0.33$ | $42.51 \pm 0.78$ | $80.90 \pm 0.15$ | $\mathbf{40.05 \pm 0.14}$ | $53.49 \pm 2.54$ |
| $\rightarrow \Delta$ DefNTaxS | +0.28 | +0.06 | -1.68 | -3.32 | -0.20 | +0.71 | -2.50 |
| TaxCLIP | $61.89 \pm 0.12$ | $53.34 \pm 0.18$ | $85.81 \pm 0.25$ | $39.28 \pm 0.59$ | $\mathbf{81.02 \pm 0.10}$ | $37.59 \pm 0.12$ | $52.46 \pm 2.27$ |
| $\rightarrow \Delta$ DefNTaxS | -1.07 | -0.25 | -0.37 | -6.55 | -0.08 | -1.76 | -3.53 |

Table 4: Comparison of DefNTaxS and random character/word-augmented approaches (WaffleTaxS and TaxCLIP) for zero-shot visual classification performance across 5 iterations, displaying the mean and standard error.

To further investigate this, we compare DefNTaxS with two random character-augmented approaches, WaffleTaxS and TaxCLIP:

- WaffleTaxS: Substitutes DefNTaxS taxonomic subcategory labels with random characters, while retaining the class descriptor.
- TaxCLIP: substitutes DefNTaxS class descriptors with random characters, while retaining the taxonomic subcategory labels.

We see mixed results across the datasets, with DefNTaxS outperforming WaffleTaxS and TaxCLIP in a significant number of cases. As the random characters replace a different element of semantic content in each approach, it may suggest that where WaffleTaxS dominates, fine-grained differentiation is the most impactful, while additional semantic contextualization is needed in case where TaxCLIP leads.

The random characters are also inserted at a different position in each approach, potentially impacting accuracy through varying weighting of earlier versus later tokens in the context window. This structural concept is investigated further in Appendix C.

As with WaffleCLIP, we observe that the ability to differentiate between classes is crucial for performance, even without interpretable semantic content. This suggests that the semantic content of the descriptors is not the only factor contributing classification, but differentiation alone has an effect.

### 6.2 LLM CLUSTERING VS TRADITIONAL CLUSTERING

To compare the effectiveness of LLMs in clustering tasks, we compare the performance of a traditional clustering approach using k-means with a CLIP-based approach. We use the same dataset as in the previous section, with the same CLIP backbones and descriptors. The only way the method differs is that the class text embeddings are provided to the k-means algorithm instead of the LLM, while the LLM still generates the subcategory labels. Results of this comparison can be found in Table 5, with a clear distinction between the two methods. The consistent performance of DefNTaxS across all CLIP backbones demonstrates the effectiveness of the LLM in generating subcategory labels. We expect this is due to the high dimensional embedding space of the CLIP backbones, which

allows for better separation of the subcategory labels, where a small, simple k-means approach would struggle to differentiate between the classes.

|         | IN    | CUB   | Pets  | DTD   | Food  | Places | ESAT  | Mean  |
|---------|-------|-------|-------|-------|-------|--------|-------|-------|
| k-means | 62.61 | 53.37 | 85.17 | 45.87 | 80.76 | 39.65  | 54.03 | 60.21 |
| LLM     | **63.48** | **54.00** | **86.09** | **45.89** | **81.22** | **40.00** | **57.22** | **61.13** |
| Δ       | +0.87 | +0.64 | +0.92 | +0.02 | +0.47 | +0.35  | +3.19 | +0.92 |

Table 5: Comparison of zero-shot visual classification performance across different benchmarks using k-means vs. LLM clustering with multiple CLIP backbones.

## 7 CONCLUSION

This work establishes taxonomic context as a fundamental requirement for robust zero-shot image classification. Through DefNTaxS, we demonstrate that the persistent ambiguity problem in vision-language models can be systematically addressed by automatically discovering and integrating taxonomic relationships into class representations.

Our key insight - that effective disambiguation requires both descriptive content and relational context - challenges the prevailing focus on isolated descriptors and hierarchical structures. DefNTaxS operationalizes this insight through fully automated taxonomic discovery, achieving consistent state-of-the-art performance across seven benchmarks with particularly dramatic improvements on ambiguity-prone datasets.

Beyond the immediate performance gains (5.5% average, 13.0% maximum), DefNTaxS represents a paradigm shift toward context-aware zero-shot learning. By eliminating the need for manual disambiguation while providing principled solutions to semantic ambiguity, our approach opens new directions for scalable, robust vision-language understanding. Future work should explore how taxonomic contextualization can enhance other vision-language tasks and how dynamic taxonomy adaptation might handle evolving class distributions in real-world applications.

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

# A  SUBCATEGORY GENERATION PROCESSES

Below are the prompts and a walkthrough of the process of generating subcategories from classes.

## A.1  GENERATING SUBCATEGORIES

The first step involves generating an initial list of subcategories for the classes in the dataset. A context prompt is used to instruct the LLM to group the classes into subcategories, formatted as a Python list.

PROMPT:

The $[dataset\ name]$ dataset is constructed from $[number\ of\ classes]$. You will create at minimum $[min\ subcategories]$ subcategories to group the classes by and assign at maximum $[max\ classes\ per\ subcategory]$ of the $[dataset\ name]$ classes to each subcategory. For an example of a subcategory and its classes, a subcategory "kitchen utensil" may have the classes "fork", "knife", "can opener" and "teaspoon" assigned to it. Every class must be assigned to a subcategory, none can be missed. First, create the list of subcategories to assign these $[dataset\ name]$ classes to, in the exact form of a Python list and nothing more, and stop there before assigning the classes.

$[dataset\ name]$ classes:
$[class\ list]$

## A.2  REFINING SUBCATEGORIES

If the generated subcategories are too broad or lack specificity, they are refined to ensure better granularity. The prompt requests LLM to break down broad categories into finer ones for better differentiation among classes.

PROMPT:

The subcategories in this list are too coarse and will not differentiate the classes well. Break down the existing subcategories into more specific subcategories to better group the classes, e.g. instead of "dog" and "cat", use "terrier", "retriever", "siamese" and "persian". Use as many as needed to allow the classes to be as distinct as possible, and even removing overly broad subcategories like "dogs" and "cats". Once again, do not assign classes yet.

Subcategories:
$[subcategory\ list]$

## A.3  ASSIGNING CLASSES TO SUBCATEGORIES

In this step, each class in the dataset is assigned to the most appropriate subcategory from the refined list. LLM is instructed to select a subcategory for each class without introducing new categories.

PROMPT:

Which of the subcategories in this Python list should '$[class\ name]$' be assigned to? It must be one of the subcategories in the list, not a new one. If a class could belong to multiple subcategories, assign it to the most unique/least likely subcategory. Respond with only the subcategory name.

## B MAIN RESULTS ON FURTHER CLIP BACKBONES

In this section, we extend the core experiments of the paper to the ViT-B/16 and ViT-L/14 CLIP backbones.

### B.1 MAIN RESULTS

| Method | IN | | CUB | | Pets | | DTD | | Food | | Places | | ESAT | | Mean | |
|---|---|---|---|---|---|---|---|---|---|---|---|---|---|---|---|---|
| | B/16 | L/14 | B/16 | L/14 | B/16 | L/14 | B/16 | L/14 | B/16 | L/14 | B/16 | L/14 | B/16 | L/14 | B/16 | L/14 |
| CLIP | 64.10 | 71.55 | 56.40 | 62.98 | 80.14 | 86.82 | 44.57 | 50.74 | 84.02 | 89.87 | 38.32 | 39.04 | 46.10 | 36.83 | 59.09 | 62.55 |
| E-CLIP | 66.60 | 72.81 | 55.89 | 62.65 | 85.51 | 91.81 | 43.62 | 51.42 | 84.88 | 89.78 | 39.19 | 39.76 | 52.74 | 54.04 | 61.20 | 66.04 |
| D-CLIP | 68.05 | 75.00 | 57.49 | 64.52 | 85.58 | 91.15 | 45.51 | 54.59 | 85.55 | 90.33 | 40.55 | 40.86 | 51.95 | 49.98 | 62.10 | 66.63 |
| W-CLIP | 67.29 | 74.15 | 56.21 | 62.34 | 81.22 | 88.24 | 42.50 | 49.41 | 85.27 | 90.62 | 39.52 | 39.86 | 31.94 | 34.28 | 57.71 | 62.70 |
| W-CLIP+concepts | 67.29 | 74.07 | 56.90 | 62.55 | 86.93 | 93.71 | 42.50 | 49.41 | 86.40 | 90.87 | 40.52 | 41.02 | 41.27 | 50.01 | 60.26 | 65.95 |
| CUPL | 66.01 | 73.68 | 56.84 | 63.03 | 84.03 | 84.60 | 42.61 | 43.87 | 83.89 | 88.97 | 39.01 | 38.57 | 38.57 | 48.25 | 58.71 | 63.00 |
| CGPT-Powered | 68.03 | 74.55 | 57.81 | 63.89 | 86.16 | 91.77 | 47.18 | 56.05 | 85.39 | 90.49 | 41.02 | 41.61 | 47.60 | 46.01 | 61.88 | 66.34 |
| CHiLS | 65.73 | 72.20 | 55.46 | 62.05 | 88.10 | 91.87 | 45.43 | 55.53 | 89.02 | 92.96 | 41.06 | 41.59 | 47.16 | 59.69 | 61.71 | 67.98 |
| *DefNTaxS* | 68.03 | 75.03 | 58.15 | 63.93 | 89.31 | 92.76 | 47.38 | 52.75 | 86.61 | 91.12 | 41.09 | 41.81 | 56.51 | 60.54 | 63.87 | 68.28 |
| → Δ CLIP | +3.93 | +3.48 | +1.74 | +0.95 | +9.17 | +5.94 | +2.80 | +2.00 | +2.60 | +1.25 | +2.77 | +2.76 | +10.41 | +23.71 | +4.78 | +5.73 |
| → Δ D-CLIP | -0.02 | +0.03 | +0.66 | -0.59 | +3.73 | +1.61 | +1.86 | -1.84 | +1.07 | +0.79 | +0.53 | +0.95 | +4.56 | +10.56 | +1.77 | +1.64 |

Table 6: Results of DefNTaxs and other baselines on ViT-B/32 and ViT-L/14 CLIP backbones

We found the results of these further two backbones largely mirror the results of the experiments on the ViT-B/32 backbone, with DefNTaxS outperforming the other methods across the ViT-B/16 and ViT-L/14 backbones, achieving the highest mean accuracy across all datasets. However, we note that the performance gap between DefNTaxS and D-CLIP is smaller on these backbones compared to the ViT-B/32 backbone, especially for the ViT-L/14 backbone, indicating that the effectiveness of taxonomic definitions may vary with different CLIP architectures. The results indicate that our approach to leveraging taxonomic definitions significantly enhances classification performance, even with different CLIP backbones.

### B.2 LLM CLUSTERING VS TRADITIONAL CLUSTERING

We continue our exploration of LLM vs traditional clustering methods for grouping classes into subcategories on the ViT-B/16 and ViT-L/14 CLIP backbones. Results of this comparison can be found in Table 7. While we still observe that the LLM-grouped performance is superior on average, the mean result for ViT-B/16 has a drastically reduced margin of improvement, and class-based performance is even below the clustered performance in some cases.

| | IN | | CUB | | Pets | | DTD | | Food | | Places | | ESAT | | Mean | |
|---|---|---|---|---|---|---|---|---|---|---|---|---|---|---|---|---|
| | B/16 | L/14 | B/16 | L/14 | B/16 | L/14 | B/16 | L/14 | B/16 | L/14 | B/16 | L/14 | B/16 | L/14 | B/16 | L/14 |
| LLM | 68.09 | 75.03 | 58.15 | 63.93 | 89.31 | 92.76 | 47.38 | 52.75 | 86.09 | 90.99 | 41.09 | 41.81 | 56.51 | 59.68 | 63.80 | 68.13 |
| k-means | 67.24 | 74.18 | 57.82 | 63.44 | 88.76 | 92.45 | 47.71 | 55.50 | 85.67 | 90.52 | 40.66 | 40.62 | 57.50 | 51.41 | 63.62 | 66.87 |
| Δ | +0.85 | +0.85 | +0.33 | +0.48 | +0.55 | +0.31 | -0.34 | -2.75 | +0.42 | +0.47 | +0.42 | +1.19 | -0.99 | +8.27 | +0.18 | +1.26 |

Table 7: Comparison of zero-shot visual classification performance across different benchmarks using k-means vs. LLM clustering with multiple CLIP backbones.

## C  FURTHER PROMPT MODIFICATION INVESTIGATIONS

We analyze how prompt structure and length affect CLIP (Radford et al., 2021) classification performance. We use the CUB (Welinder et al., 2010) dataset to evaluate CLIP's sensitivity to prompt variations.

### C.1  PROMPT STRUCTURE

Table 8: Impact of Different Prompt Structures on Zero-Shot Classification Accuracy

| Method | Prompt Structure | Accuracy (%) |
|---|---|---|
| E-CLIP Baseline | "A photo of a {c}" | 51.95 |
| D-CLIP Baseline | "{c}, which is/has/etc {d}" | 52.57 |
| Class-Descriptor Switch | "{d}, which is/has/etc {c}" | 51.34 |
| Prefix Modification | "An image of a {c}, which has/is {d}" | 50.94 |
| Class-Specific Prefix Modification | "A photo of a {c}, which has/is {d}, a type of bird" | 53.33 |
| Class Label Modification | "{$c_{mod}$}, which is/has/etc {d}" | 22.14 |
| Descriptor-Only | "{d}" | 3.81 |
| Class Repetition | "{c}, which is/has/etc {c}" | 52.35 |

We analyzed D-CLIP's (Menon & Vondrick, 2023) prompt structure "$c_i$ which has/is $d_i$," where $c_i$ is the class and $d_i$ is the descriptor.

Reversing class and descriptor positions to "$d_i$, which is a description of a $c_i$" reduced accuracy, despite initial tokens having greater embedding weight (Han et al., 2024; Kazemnejad et al., 2024). This shows classes perform better at prompt start.

Adding prefixes like "An image of a $c_i$, which has/is $d_i$" decreased accuracy by shifting focus from the class. Classes performed best at prompt start without prefixes. However, domain-specific templates like "a photo of a class label, a type of bird" improved accuracy, showing that tailored templates can overcome general prefix limitations.

Simplifying class names (e.g., "Red-winged Blackbird" to "Blackbird") reduced accuracy by removing distinctive features and over-relying on descriptors.

Eliminating class names entirely and using only descriptors caused sharp accuracy decline, highlighting class labels' critical role.

Replacing descriptors with repeated class names improved accuracy, demonstrating that class labels are vital for model performance. These findings emphasize the importance of class positioning and inclusion in prompt design. The summary of all prompt structure modifications, along with their respective accuracy results, is presented in Table 8.

### C.2  PROMPT LENGTH

| Method | Accuracy (%) |
|---|---|
| CLIP Baseline | 51.95 |
| D-CLIP GPT-3 Baseline | 52.57 |
| D-CLIP GPT-4 Baseline | 53.90 |
| # random characters: 2 | 51.55 |
| # random characters: 5 | 51.87 |
| # random characters: 10 | 51.10 |
| Truncation (Class label only) | 51.78 |
| Truncation (Maximum @ 100%) | 53.88 |
| Truncation (Minimum @ 10%) | 50.77 |
| Truncation (@ 0%) | 52.23 |
| Truncation (@ 50%) | 51.34 |
| Truncation (@ 70%) | 53.59 |

Table 9: Impact of Length of Prompt on Zero-Shot Classification Accuracy

We examine prompt length's effect on classification accuracy, independent of semantic content.

First, we truncate descriptors to specific character fractions while maintaining prompt structure. For example, 20% truncation of a 100-character descriptor retains the first 20 characters. 0% truncation leaves only class labels and punctuation (e.g., "Black-footed Albatross").

Accuracy decreases with progressive truncation, minimizing at 10–20% truncation. The "class label only" prompt performs worse than minimally truncated descriptors, showing partial descriptor value.

Second, we append random strings to class labels (e.g., "Black-footed Albatross ghdf idfh") to isolate length effects without semantic changes. Table 9 presents truncation levels and corresponding accuracies.

## C.3 PROMPT REGENERATION WITH MODERN LLMS

We investigate whether regenerating descriptors with advanced LLMs (GPT-4) provides similar benefits to taxonomic refinement. More powerful LLMs may better recognize task-relevant semantic features when generating descriptors.

Table 10 shows that regenerating descriptors with advanced LLMs alone does not match taxonomic refinement benefits. However, combining both approaches significantly improves classification accuracy, indicating that subcategories and enhanced descriptors complement each other.

| Method | IN | | | CUB | | | Pets | | | DTD | | | Food | | | Places | | | ESAT | | |
|---|---|---|---|---|---|---|---|---|---|---|---|---|---|---|---|---|---|---|---|---|---|
| | B/32 | B/16 | L/14 | B/32 | B/16 | L/14 | B/32 | B/16 | L/14 | B/32 | B/16 | L/14 | B/32 | B/16 | L/14 | B/32 | B/16 | L/14 | B/32 | B/16 | L/14 |
| CLIP | 58.86 | 64.07 | 71.57 | 51.83 | 56.35 | 62.98 | 77.96 | 80.12 | 86.83 | 41.08 | 44.59 | 50.76 | 77.84 | 84.02 | 89.86 | 37.48 | 38.33 | 39.05 | 44.32 | 46.20 | 36.97 |
| E-CLIP | 61.90 | 66.61 | 72.80 | 51.95 | 55.87 | 62.70 | 82.06 | 85.49 | 91.87 | 43.12 | 43.60 | 51.44 | 78.79 | 84.86 | 89.78 | 39.12 | 39.18 | 39.75 | 33.31 | 52.56 | 54.10 |
| CuPL | 62.10 | 67.23 | 73.31 | 51.97 | 56.89 | 63.45 | 80.89 | 82.25 | 90.78 | 45.51 | 45.95 | 53.61 | 79.26 | 83.71 | 90.02 | 39.84 | 40.55 | 40.86 | 47.36 | 51.95 | 49.98 |
| D-CLIP | 63.26 | 68.38 | 75.16 | 53.83 | 59.13 | 65.34 | 81.54 | 85.64 | 91.58 | 47.11 | 47.64 | 56.54 | 81.06 | 86.09 | 91.22 | 40.89 | 41.85 | 41.46 | 42.80 | 49.85 | 46.08 |
| W-CLIP | 62.26 | 67.18 | 74.11 | 52.05 | 55.42 | 62.75 | 80.12 | 81.27 | 88.17 | 41.03 | 44.49 | 50.46 | 80.31 | 85.23 | 90.60 | 38.64 | 39.64 | 40.10 | 35.28 | 49.47 | 46.24 |
| W-CLIP+concepts | 62.26 | 67.18 | 74.11 | 52.49 | 56.18 | 63.13 | 85.33 | 86.64 | 93.88 | 41.03 | 44.49 | 50.46 | 81.56 | 86.41 | 91.28 | 40.62 | 40.82 | 41.25 | 46.05 | 49.39 | 51.59 |
| DefNTaxS | 63.63 | 68.28 | 75.05 | 54.42 | 59.53 | 64.62 | 86.67 | 89.27 | 93.14 | 48.09 | 48.87 | 54.57 | 81.47 | 86.45 | 91.28 | 39.31 | 40.81 | 40.96 | 57.51 | 60.25 | 60.67 |

Table 10: Comparison of zero-shot visual classification performance across different image classification benchmarks using multiple CLIP backbones and descriptors generated by GPT-4.

## C.4 WAFFLETAXS AND TAXCLIP ORDER SWAPS

| Method | IN | CUB | Pets | DTD | Food | Places | ESAT |
|---|---|---|---|---|---|---|---|
| W-TaxS$_{swap}$ | **63.26 ± 0.07** | **53.78 ± 0.18** | 84.49 ± 0.36 | 40.43 ± 1.07 | 81.02 ± 0.04 | 39.78 ± 0.16 | 52.22 ± 2.34 |
| ⇀ Δ DefNTaxS | +0.30 | +0.20 | -1.68 | -5.39 | -0.08 | +0.44 | -3.77 |
| ⇀ Δ W-TaxS | +0.02 | +0.13 | -0.00 | -2.08 | +0.12 | -0.27 | -1.28 |
| TaxCLIP$_{swap}$ | 61.18 ± 0.12 | 53.50 ± 0.22 | 85.43 ± 0.34 | 39.37 ± 0.37 | **81.06 ± 0.10** | 38.26 ± 0.15 | 52.48 ± 2.37 |
| ⇀ Δ DefNTaxS | -1.78 | -0.09 | -0.74 | -6.46 | -0.04 | -1.08 | -3.51 |
| ⇀ Δ TaxCLIP | -0.71 | +0.16 | -0.37 | +0.09 | +0.05 | +0.67 | +0.02 |

Table 11: Comparison of DefNTaxS and random character/word-augmented approaches, Waffle-TaxS, TaxCLIP, WaffleTaxS$_{swap}$ and TaxCLIP$_{swap}$, for zero-shot visual classification performance across 5 iterations, displaying the mean and standard error.

Extending on Section 6.1.3, we also introduce a "swapped" variant of each WaffleTaxS and TaxCLIP, where the order of the class descriptor and subcategory label are swapped in the prompt, e.g. "*forks*, which are commonly found among *utensils*, and which has *tines*".

Except for a few notable but difficult to reconcile exceptions, results roughly resemble though of vanilla WaffleTaxS and TaxCLIP.

## D INVESTIGATION OF SUBCATEGORY SIZE

In this section, we investigate the impact of the number of classes assigned to each subcategory on the performance of our proposed method, DefNTaxS. We analyze how varying the number of examples in each subcategory affects the overall classification accuracy.

When defining size in this context, we refer both to the number of classes a subcategory is constrained to and the number the LLM is prompted to assign to each subcategory.

We observe that the performance of DefNTaxS is sensitive to the number of classes assigned to each subcategory, with a clear trend of performance improvement as the number of classes per subcategory increases, tapering off at around 20 classes per subcategory. For this reason, we selected a maximum of 20 classes per subcategory for a balance between performance and computational efficiency.

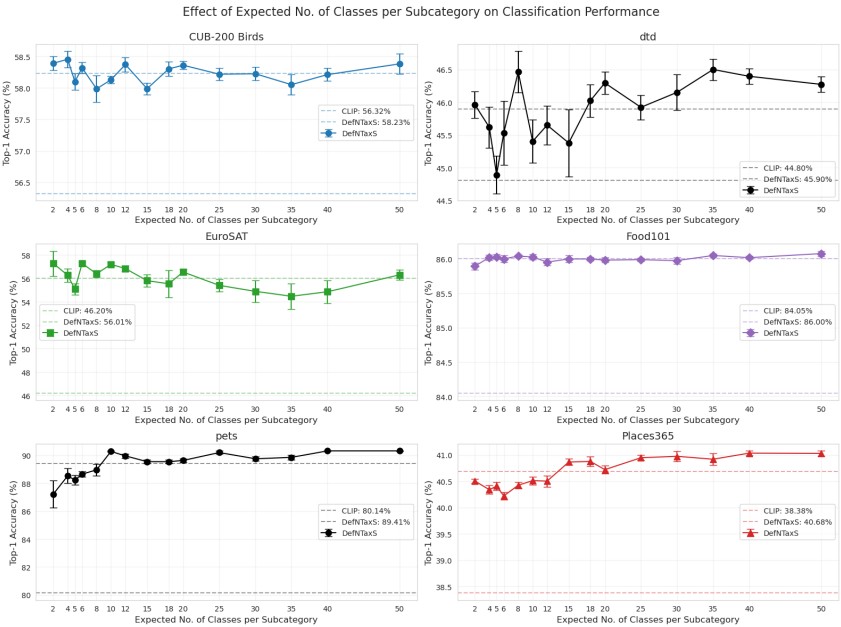

Figure 2: The impact of the number of classes assigned to each subcategory on the performance of DefNTaxS. The effect varies across datasets, with some showing a clear trend of performance improvement as the number of classes per subcategory increases, while others show less sensitivity to this parameter. In the cases of improvement, we see a stabilization of performance at around 20 classes per subcategory, suggesting this is an upper limit for the class count to assign to each subcategory.

# E    SUBCATEGORY RANDOMIZATION

To confirm that performance is proportional to the accuracy of the subcategories to each classes, we randomize both the subcategory assignment processes in multiple ways.

## E.1    RANDOMIZATION OF ACCURATELY GENERATED SUBCATEGORIES

First, we generate the list of subcategories as described in A.1 and A.2 and assign them as normal, but then sample a percentage of the subcategory assignments to randomly assign them to a different subcategory. This approach can be observed in 3, in which we can clearly observe a trend of performance decreasing as the percentage of randomization increases. It is important to note that the performance often does not drop below the CLIP baseline, which indicates that the subcategories are still somewhat useful, likely due to their role as differentiators between classes.

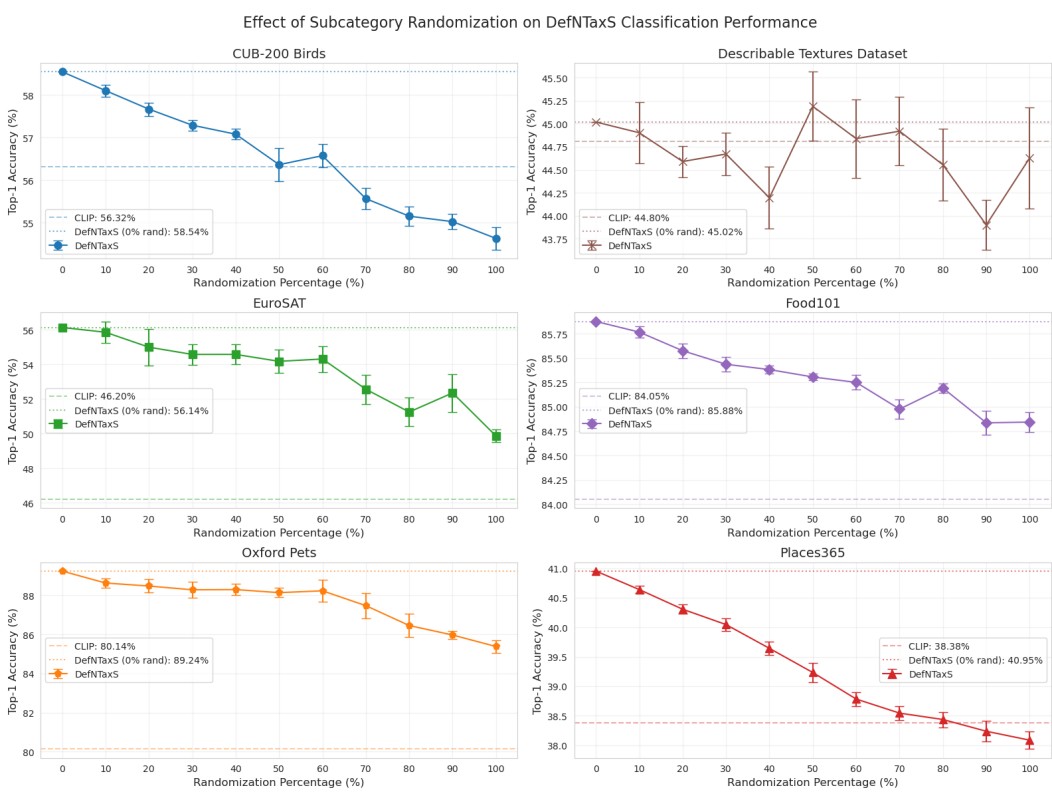

Figure 3: As the percentage of randomization increases, the performance of the model on the dataset with randomized subcategories decreases. The performance is compared to the original, more accurate subcategories and the CLIP baseline.

## E.2    RANDOM GROUPING OF CLASSES TO FORM SUBCATEGORIES

In the second approach, we randomly group the classes into subcategories without any consideration for their actual relationships and then assign names to these subcategories. In both cases, we then evaluate the performance of the model on the dataset with these randomized subcategories and compare it to the performance with the original, more accurate subcategories. In this approach, we observe a surprisingly high performance as compared to the significant drop in performance seen with the first approach, as shown in 12. However, we do also see a significantly increased variance in the performance, making it hard to draw clear conclusions about the effectiveness or lack thereof of the randomized subcategories.

| Method | CUB B/16 | Pets B/16 | DTD B/16 | Food B/16 | Places B/16 | SAT B/16 |
|---|---|---|---|---|---|---|
| DefNTaxS | **59.53 ± 0.45** | **88.95 ± 0.28** | **46.63 ± 0.25** | **86.09 ± 0.04** | **40.77 ± 0.22** | **56.94 ± 0.92** |
| DefNTaxS$_{rand}$ | 58.21 ± 0.20 | **88.64 ± 0.89** | **45.21 ± 0.93** | 85.68 ± 0.26 | 39.91 ± 0.25 | 52.74 ± 2.59 |

Table 12: By repeatedly randomly grouping the classes into subcategories and assigning names to them across 10 iterations, we can observe the performance of the model on the dataset with these randomized subcategories. The performance is compared to the original, more accurate subcategories. Results are bold when their performance is the best or within the range of the best result. ImageNet has been omitted from this investigation due to API rate limiting caused by the scale of the dataset.

## F CONCEPTUAL VISUALIZATION OF EFFECT OF SUBCATEGORIES

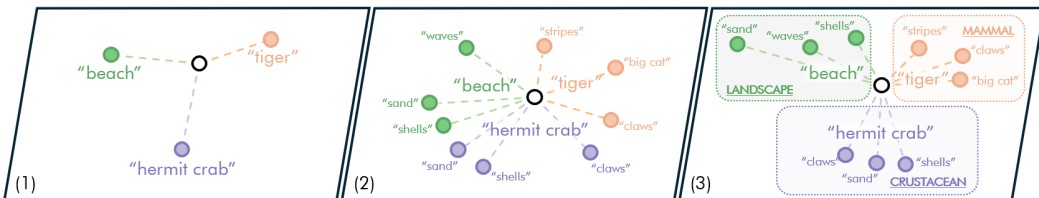

Figure 4: [Derived from Menon & Vondrick (2023)] (1) CLIP classifies an image as the class with text embedding (colored dots) that has highest cosine similarity to the target image embedding (white dot). (2) Methods like D-CLIP introduce visual descriptions to augment the text embeddings, but this can introduce ambiguities (e.g. descriptors for crabs and tigers both contain "claws", reducing separation in embedding space). (3) DefNTaxS contextualizes both the class and its descriptors, minimizing ambiguity to enhance inter-class differentiation, e.g. embeddings containing "claws", "sand", and "shells" are now separated.

