# OpenReview forum: "DefNTaxS: The Inevitable Need for Context in Classification"
_ICLR.cc/2026/Conference — Submitted to ICLR 2026_

### Official Review · Reviewer_SrrG · 2025-10-31

**Soundness:** 3
**Presentation:** 2
**Contribution:** 2
**Rating:** 6
**Confidence:** 3

**Summary:**

This paper organically integrates two lines of zero-shot CLIP enhancement methods—text prompt augmentation and hierarchy-based approaches—into a unified workflow to improve CLIP’s zero-shot classification performance.

**Strengths:**

The proposed method demonstrates a clear and rigorous workflow, supported by significant and well-validated experimental results.

**Weaknesses:**

The performance of the proposed method may potentially depend on the class structure of the classification task.

Although the paper emphasizes that taxonomic context is essential for classification, there is limited analysis supporting this claim beyond the reported accuracy of the proposed method. In Table 4, removing the taxonomic context (W-TaxS) outperforms removing the descriptors (TaxCLIP) in several cases, which does not appear to substantiate the assertion that taxonomic context is essential.

**Questions:**

Could you please include additional experimental results comparing the proposed method with existing baselines on fine-grained benchmarks, such as Oxford Flowers 102 and Stanford Cars?

---

> ### Author Response · Authors · 2025-11-18
> **Discussion around "taxonomic context" context, and datasets extension w/ timeline**
>
> - How “essential” is taxonomic context
>     - To keep consistent with how we aimed to use the phrase “taxonomic context”, we specifically mean that descriptors need to be given full *context* through taxonomic labels to be most effective, while not making claims that taxonomic labels will do better than descriptors in isolation (as in D-CLIP). The largely consistent performance improvements over D-CLIP aren’t the only part of our argument, no, but the augmentations throughout Section 6.1 emphasise the benefits of the taxonomic context, even in the scenarios where the role it plays is simply as an inter-class differentiator.
>     - The experiments involving W-TaxS and TaxCLIP are not necessarily counterexamples to this statement, but instead aim to explore whether this benefit is more so due to additional semantic information or non-semantic differentiation, as it could be argued all classification requires both knowing what something is and also having something to differentiate it from similar things, even if that’s just noise. We also ran a set of experiments where we also swapped the order of the descriptors and subcategories in the text input (Appendix C.4), which suggests some of the observed effects in Table 5 may be due to the benefits of noise as a differentiator, i.e. the datasets that perform better on WaffleTaxS perform even better when the noise is earlier in the text and having more effect due to earlier positional encoding. (Note: this was separated due to page limitations in a previous version, but we understand it makes it difficult to analyse overall and will consider joining it to Table 4 again upon revision)
> - Results for Oxford Flowers and Stanford Cars
>     - We’ll aim to test on these two datasets for completeness (with a relevant D-CLIP baseline using the method in Menon et al 2023) with the aim to have them up by the 23rd, but will also post a follow-up comment when we’ve uploaded any new results. In the meantime, we highlight that CUB is also considered a fine-grained benchmark, consisting of birds of the same genus that often share similar physical descriptors.
>     - EDIT: upon review, we recognise the potential difficulty of recreating even these two datasets across 9 (mostly distinct) baselines within the discussion period. While adapting the code for some will be trivial, we aren't so certain this will be case for all. We will still aim to and provide updates for any further results.

---

> > ### Comment · Reviewer_SrrG · 2025-11-26
> >
> > Thanks for your response.
> >
> > > How “essential” is taxonomic context
> >
> > Thank you for your clarification. I agree that context is important and does provide benefits beyond descriptors. I believe that additional insights and analysis on the differences in mechanism between context and descriptors could further strengthen the claim that context is essential and enhance the overall contribution of this work.
> >
> > > Results for Oxford Flowers and Stanford Cars
> >
> > I understand the difficulties related to the timeline.

---

> > > ### Author Response · Authors · 2025-12-03
> > >
> > > Thanks again for your time, especially across a surprisingly dramatic review period!
> > >
> > > > I believe that additional insights and analysis
> > >
> > > Do you have specific experiments you would like to see to support this? Multiple experiments contrast the use of descriptors in isolation and in context, as well as the contextualizing information used in isolation. See:
> > > - Results (D-CLIP vs DefNTaxS), further in Appendix B1
> > > - Section 6.1.2 (DefNTaxS no desc.)
> > > - Related explorations in Appendix C
> > >
> > > For more intuition around the mechanism at play here, let's refer to how the CLIP encoder handles positional encoding. As discussed in LM-Infinite (Han et al., 2024), the CLIP positional encoding strongly uses early tokens to allocate absolute position, with each following token being embedded geometrically relative to the previous positions. While we can't mechanistically understand every exact position where each concept is placed within the CLIP embedding space, we at least know that this steady building of concepts applies this relative construction method to leverage the context.
> > >
> > > Please let us know if there is further experimentation needed and what would be expected to meet this expectation.
> > >
> > > 1. Chi Han, Qifan Wang, Hao Peng, Wenhan Xiong, Yu Chen, Heng Ji, and Sinong Wang. LM-Infinite:
> > > Zero-shot extreme length generalization for large language models. In Proceedings of the 2024
> > > Conference of the North American Chapter of the Association for Computational Linguistics:
> > > Human Language Technologies (Volume 1: Long Papers), pp. 3991–4008, 2024.
> > >
> > > > difficulties related to timeline
> > >
> > > Apologies, due to other conference overlaps, issues with and responses to the OpenReview leak, and originally noted difficulties of the task, we won't be able to complete adapting all baselines to these extra datasets. All baselines previously used (most of) the existing datasets, so little adaptation was needed. If the key issue was fine-grained information though, the CUB dataset is explicitly a fine-grained bird identification benchmark to give this level of insight.

---

### Official Review · Reviewer_HnWT · 2025-10-31

**Soundness:** 2
**Presentation:** 2
**Contribution:** 2
**Rating:** 2
**Confidence:** 4

**Summary:**

This paper presents a simple approach for prompt adaptation to improve zero-shot classification of Vision Language models such as CLIP. The authors propose constructing prompts with supercategories appended to prompts for each class. Evaluation is conducted over multiple benchmarks. Ablations and improvements over baselines is shown.

**Strengths:**

The authors show improvement over baselines for most datasets. Extensive evaluation includes performance of the method over various CLIP architecures as well as ablations over different aspects such as prompt construction choices.

**Weaknesses:**

Novelty is very limited. There is very little difference conceptually wrt CHiLS and CGPT-P (cited in the paper). Why have the authors decided to only include one level in their hierarchical tree? This has not been discussed. What happens if you construct a deeper tree? This also brings into concern the fact that simply appending the super category name to the class prompt should be suboptimal as CLIP often suffers with long context without specialized fine-tuning. The performance reported does not match numbers in other papers like CHiLS for some datasets. What could cause this? Table 1 does not state the numbers are for which CLIP architecture. The lack of examples of what the texts look like also leads to confusion about the reason why this approach will work better than CHiLS and CGPT-P.

**Questions:**

Please refer to questions in weaknesses.

---

> ### Author Response · Authors · 2025-11-19
> **Expand on reductive conceptual understanding + clarify/fix technical concerns**
>
> Thank you for the time and effort taken to engage with our work, and the respect it shows for the time we put into it.
>
> Concept novelty:
>
> While all three methods incorporate hierarchical information, its insultingly reductive to suggest that’s all these sophisticated methods are. For comparison:
>
> **CHiLS**
> 1. Begin with dataset of broad (super)classes, as fine-grained datasets often can’t be refined further
> 2. Use LLM to map superclasses to generated subclasses, e.g. (superclass) “dog” → (subclasses) “shiba inu”, “pug”, etc
> 3. Calculate CLIP probabilities as the cosine similarity of image and subclass text embeddings
> 4. Map the argmax of the embedding probabilities to the superclass it was refined from. This is taken as the overall prediction for the target image
> Key note:
> - Each pair of subclass and superclass labels remain independent
>
> **CGPT-P**
> 1. Begin with any dataset of classes
> 2. Form the hierarchy:
>     1. Instruct LLM to generate class descriptions that emphasise inter-class comparisons (often non-visual), e.g. “webbed feet for swimming and water rescue”, “gentle and friendly expression with drooping jowls”
>     2. CLIP embeds the descriptor text
>     3. k-means embeddings clusters define groups
>     4. Repeat to build an arbitrary number of leaf nodes
> 3. Evaluate a complicated scoring function of the weighted average between (1) the standard CLIP prediction for a class and (2) the running average of the longest sequence of monotonically increasing values of cosine similarity between the image and leaf node category descriptions
>     1. The function in (2) is also highly parameterized, with the parameters explicitly selected to maximize results with no theoretical justification or tuning method
> 4. The predicted class is the base of the highest scoring series of nodes
> Key notes:
> - Every class has as many text embeddings approximating it as there are layers in the knowledge tree
> - The multiple layers of the knowledge tree, use of a running average, and careful hyperparameter selection indicate CGPT-P’s classification mechanism is more comparable to that of a progressive filter than a standard classifier
>
> **DefNTaxS**
> 1. Begin with any dataset of classes
> 2. An LLM is instructed to generate a set of descriptors for each class, as per D-CLIP. The LLM also creates a set of subcategories within the overall set of classes.
> 3. The LLM iteratively assigns individual classes to each subcategory. If too many are assigned to a single subcategory, it is deemed to be a poor differentiator and step 2 is repeated for that subcategory, e.g. “dogs” → “big dogs”, “farm dogs”, “fluffy dogs”
> 4. Embed text prompts of the form shown in Section 3.5
> 5. Calculate the prediction as the argmax of the average of the CLIP probabilities for each class prompt
> Key note:
> - A small number of text prompts directly represent the class, with the class label grounding positional encodings
>
> **Summary**: we observe abunantly clear variation in applicable dataset regimes, hierarchy depth and generation methods, probability calculations and scoring methods, mappings (or lack thereof), and intrinsic hyperparameter dependence. Claiming a lack of distinction between these methodologies displays a shallow or poor understanding of some or all.
>
> Hierarchical depth vs CLIP limits:
> - We considered both issues of further taxonomic context and CLIP context limitations (the latter discussed at the end of Section 6.1.2). Long-CLIP (Zhang et al, 2024) suggests the effective CLIP context window is ~20 tokens (much shorter than its 77 token max), with performance flattening beyond this. Our text inputs mostly fall comfortably in the lower 20-30 token range. However, we did run naïve tests extending to deeper taxonomic context, but many prompts significantly passed this range and performance dropped.
>
> Inconsistency of results with other papers:
> - Models commonly experience output variations due to changes in datasets across versions, as well as software (esp. CUDA) / package versions on different machines. To set a consistent control, we recreated each baseline locally from the code provided. We expected some deviation from original figures, but changes were often negligibly small or even stronger relative to other baselines than originally reported, e.g. CGPT-P on Pets on ViT-B/32.
>
> Backbone used in main results:
> - The CLIP backbone used for all experiments in the main paper was ViT-B/32. We apologise, this was previously included in 4.1 but must have inadvertently been removed. We’ll fix this in our next version soon, alongside changes requested by other reviewers and a change log. Thank you for picking that up.
>
> Examples:
> - To clarify, are you requesting examples of the CHiLS and CGPT-P text inputs? As noted, the specific text prompts aren’t the primary distinction between methods, but examples of our texts can be found in Section 3.4, with pipeline-contextualized samples of the others above. We also suggest Fig 1 in CHiLS +Fig 4 in CGPT-P.

---

### Official Review · Reviewer_ndRT · 2025-11-02

**Soundness:** 2
**Presentation:** 2
**Contribution:** 2
**Rating:** 2
**Confidence:** 4

**Summary:**

The paper introduces DefNTaxS, which addresses taxonomic ambiguity in zero-shot classification by automatically generating LLM-based taxonomic context for class prompts. Building on D-CLIP's approach of adding descriptors to class names, DefNTaxS extends the template to include subcategory information: while D-CLIP uses "[class] which [has/is] [descriptor]", DefNTaxS adds taxonomic context as "[class] which [has/is] [descriptor], [contextual phrase] [subcategory]". The method uses GPT-X to discover subcategories, assign classes (targeting ~20 classes per group), and generate natural contextual phrases, achieving +5.5% average improvement over CLIP and +2.44% over D-CLIP across seven benchmarks without requiring model retraining.

**Strengths:**

The work explores an interesting aspect of vision-text alignment by investigating how taxonomic context can help resolve semantic ambiguities in CLIP-style models, addressing a real limitation where classes like "boxer" could refer to dogs or athletes depending on dataset context.

**Weaknesses:**

-- marginal extension over existing work: The contribution is an incremental improvement over D-CLIP and related LLM-prompting methods (WaffleCLIP, CuPL, CHiLS), essentially adding one more field to existing prompt templates. The taxonomic discovery process reduces to basic LLM prompting with heuristic post-processing rules, offering limited technical novelty beyond existing descriptor-based approaches.

-- small performance gains: The improvements over D-CLIP are modest (+2.44% average, table 1) and inconsistent across datasets, with some showing minimal gains (+0.16% on Places365) or even degradation (-1.05% on Food101).

-- limited technical and scientific contribution: the paper lacks architectural insights about vision-language models and proposes no learnable components or model modifications. The study is mainly about prompt manipulation, which doesn't advance our understanding of why CLIP struggles with ambiguity or how to fundamentally improve VLM architectures.

**Questions:**

--

---

> ### Author Response · Authors · 2025-11-18
> **Rebuttal to what largely seems dismissive of zero-shot approaches**
>
> - Novelty
>     - We should highlight that CHiLS is not a prompting-based method, and CuPL significantly varies from the structured methods of WaffleCLIP, D-CLIP, and our DefNTaxS. If our findings around semantic vs non-semantic differentiation (and the methods built around them) offer no insights past other methods, please explain the consistent improvements across all relevant datasets, as well as the consistent observable effects of refinement of subcategories (6.1.1) and lack of performance of other modifications of semantic information (6.1.2, Appendix C.1-C.3)
> - Performance
>     - While we disagree with the overall sentiment - when operating in a regime that prevents the use of further data, even a small improvement should be considered of significance - we’re mostly unsure where degradation is shown between Food101 on the D-CLIP baseline (80.43%) vs the DefNTaxS method (81.48%). The “+” sign there may have been read as a “-” sign, but we have also highlighted improvements in green to aim to make the change in performance clear for the human eye.
> - Scientific value
>     - Respectfully, there are many questions about CLIP and other VLMs that this work doesn’t answer in their entirety, but these questions were also not the aim of this study. We aimed to expand on the understanding of whether semantics were the key element for performance in VLM-based zero-shot classification methods. Where WaffleCLIP made claims that VLMs simply can not handle fine-grained semantics, we have shown that their work actually discovered the role of even non-semantic differentiation in disambiguating similar classes. We built on and gave new perspectives to these findings, creating a method that leverages these non-semantic differentiation factors while also boosting the ability to leverage semantics present in the text.
>     - Further, the field of zero-shot classification largely focuses on lightweight methods for improving performance of a given model, prohibitively avoiding learnable components or modifications (otherwise it wouldn’t be zero-shot). We reject the notion that because our work is in the field of zero-shot classification rather than alternative VLM architectures (for example), it therefore can not provide valuable insights to others seeking to apply these methods to lightweight VLM architectures.

---

### Official Review · Reviewer_dXFM · 2025-11-09

**Soundness:** 2
**Presentation:** 3
**Contribution:** 2
**Rating:** 2
**Confidence:** 4

**Summary:**

This paper extends existing works on image-text matching within VLMs such as CLIP by augmenting class descriptors with taxonomical context, achieving a gain of on average 5.5% over CLIP on standard benchmarks.

**Strengths:**

The paper is well written and simple to follow, and on standard benchmarks compared to methods such as D-CLIP or WaffleCLIP, consistent performance improvements can be found.

**Weaknesses:**

I have several issues with the proposed setup, and would love for the authors to provide more context here:
* Why would DEFNTAXS actually be a meaningful improvement over D-CLIP, which itself has significant issues, as descriptors proposed by LLMs are not necessary to be found in given query images, see Roth et al. 2023? The problems are the same, it's just that more context is provided that may or may not match the query image, no? I.e., adding "commonly found in kitchen utils" would maybe help to differentate a fork against, say, a motorcycle, but the fail when contrasting between potentially a single fork, and a general image on kitchen utensils, since redundancy is introduced.
* Importantly, a lot of additional semantic information is introduced into a language encoder that is known to be very weak and often fails to distinguish finegrained or multiples of semantic features. It would be important for the authors to offer some support that the CLIP model actually meaningfully and explicitly leverages the semantic context; and its not just better chosen structured noise which is given the additional performance gains.

**Questions:**

See weaknesses.

---

> ### Author Response · Authors · 2025-11-18
> **Great engagement with the core concepts, well appreciated review**
>
> Thanks for the time and effort you’ve put into the review and really trying to get into the underlying concepts with us, we always appreciate it. Apologies in advance for the long responses, but we wanted to make sure we’re thorough with the limited discussion opportunity available. We hope our labels (W1 = weakness 1) are clear in relation to your questions.
>
> - W1 (general concept):
>     - This is a good question, we’re glad you brought it up. Regarding the core issue that covers D-CLIP as well, it’s a common and often implicit assumption that the text inputs to VLMs are directly correlated to visual elements of the image inputs. While there are methods that aim to formally enforce this (e.g. Grad-CAM), majority of the time this relies solely on the idea that models work like we do: see the picture, break it down, and compare specific elements to our knowledge base of previous experiences/”data”.
>     - However, findings from works like “If CLIP Could Talk” (Esfandiarpoor et al, 2024) suggest that visual descriptions outperform non-visual descriptions only about ~20% of the time! This is less surprising when we remind ourselves these models simply work based on their training data, and internet-based data very rarely describes in detail the individual elements of an image. Both our work and D-CLIP’s may seem less interesting overall if we were able to adjust your base perspective to this: the best method from zero-shot classification using LLM-generated prompts comes when the LLM creates text similar to that found alongside this image type in the training data.
>     - While we didn’t base our work off of theirs, further results from Esfandiarpoor et al. 2024 found this included text content about habitats, behaviour, timing, and even statements like “CLICK HERE TO ENLARGE PHOTO OF <class>”. From this perspective, all we expect we’ve done is created a method that more consistently recreates image captions from, say, Wikipedia. While maybe not intellectually impressive, it does have novelty, consistency, and is scientifically aligned with our existing understanding of machine learning as a whole rather than relying on some form of unexplainable notion of reasoning.
> - W1 (specific example):
>     - For the example of forks vs general kitchen utensils, that may be the case, yes, but we would also note the poor dataset design if overlapping classes were defined like this. Further, in the case where you’re attempting to classify an image as “fork” when it shows a pile of forks, spoons, knives, etc (with all being valid classes in the dataset), we’re not sure there’s any method that could conclusively classify the “correct” answer where they’re all technically correct.
>     - The key factor we may rely on is the positional encoding of the target class at the beginning of the text input, leading to the embedding being located closer to “fork” than any other in embedding space. For further on this, refer to Factor 3 in LM-Infinite (Han et al., 2024), as well as our empirical investigations on reordering of prompts (and other modifications) in Appendix C.1. It’s surprising and counterintuitive how much a modification may be seemingly semantically benign to you and I, but may still have significant effects on VLMs (e.g. simply repeating the class name in place of D-CLIP descriptors drops D-CLIP performance by only 0.22%, whereas switching the order of the class and descriptor in the prompt loses 1.23%)
> - W2:
>     - This is the purpose of the investigation in 6.1.3 (and less so in Appendix C). In both WaffleTaxS and TaxCLIP, we replace some element of semantic information with text noise (random characters) to examine their effects. In some cases, WaffleTaxS actually performs better than DefNTaxS, suggesting that semantics may actually help less than noise in situations with overlapping class attributes (e.g. in CUB) or huge numbers of classes (e.g. ImageNet). In these situations, it is expected that the model may have a better concept of certain classes like “Grey Albatross” and the distinction from a “Black Albatross” than can be described with discrete semantic phrases (as with traditional image classifier models), so including them may simply serve confuse the model.
>     - The key strength for DefNTaxS: a balance of the benefits of semantics and noise are found in the ensembling of multiple semantic descriptors and averaging the results of each. While the benefits of noise would be expected to be equal for each individual case, the semantics are unique to each. See also Appendix E where we randomly shuffle/assign subcategories and observe consistently reducing performance with more randomization of subcategory labels, which we would not see if the core results were due to pure noise.

---

### Meta-Review · Area_Chair_ViT1 · 2026-01-06

[review text omitted: it was posted to a different submission]

---

### Decision · Program_Chairs · 2026-01-26

Reject